# HGR-TabE: Universal Tabular Embeddings via Maximal Correlation Alignment

**Niharika S. DSouza** [1]  **Liane Vogel** [2]  **Kavitha Srinivas** [3]  **Sola Shirai** [3]  **Oktie Hassanzadeh** [3]  **Horst Samulowitz** [3]

## Abstract

Universal text embedding models show that a single pretrained model can produce representations useful across tasks like classification, clustering, and retrieval. In contrast, tabular foundation models remain largely task-specific. We ask whether a single tabular embedding model can generalize across tasks. We propose HGR-TabE, an initial approach that first aligns heterogeneous table-cell representations into a shared space using Hirschfeld–Gebelein–Rényi (HGR) maximal correlation, capturing relationships between numerical and non-numerical values within rows. We then apply message passing via Hypergraph Transformer (All-Set Transformer modules) to preserve row and column permutation invariance. The model is trained entirely with self-supervision to learn consistent representations at the cell, row, column, and table levels. Without task-specific fine-tuning, it generates embeddings that perform well on row similarity, column similarity, and predictive tasks, demonstrating strong cross-task generalization compared to specialized models.

## 1. Introduction

Universal text embedding models demonstrate that a single pretrained model can support diverse tasks such as classification, clustering, and retrieval, as exemplified by MTEB (Muennighoff et al., 2023). In contrast, tabular foundation models remain largely task-specific.

*Limitations of existing approaches.* Recent methods (Qu et al., 2025; Hollmann et al., 2025; Kim et al., 2025) leverage large-scale pretraining to improve predictive performance, but focus primarily on supervised learning rather than producing reusable embeddings. Similarly, task-specific systems for table retrieval or QA (Khatiwada et al.,

2025; Herzig et al., 2021) do not generalize across tasks. This motivates the question this paper asks: can a **single tabular embedding model** support both table understanding and prediction?

*Challenges.* Tabular data is inherently heterogeneous, combining numerical, categorical, temporal, and textual features with context-dependent semantics. Purely numerical approaches (Hollmann et al., 2025) are effective for prediction but discard semantic information crucial for tasks such as entity matching and column similarity. Conversely, applying language models directly to tables is problematic: linearization obscures structure, introduces spurious dependencies, and poorly captures numerical relationships (Hegselmann et al., 2023; Fang et al., 2024; Thawani et al., 2021).

*Our approach.* We propose HGR-TabE, a tabular embedding model that learns unified representations across cells, rows, and columns. The model aligns heterogeneous cell values within rows and aggregates them across table structure, enabling a single embedding space usable across tasks. Training is fully self-supervised, i.e. label-free.

*Contributions.* (1) We introduce a unified tabular embedding framework that supports both table understanding and prediction. (2) We design alignment and aggregation mechanisms to handle heterogeneous data while preserving table structure. (3) We demonstrate good performance across row similarity, column similarity, and predictive tasks, showing improved cross-task generalization over specialized language models and tabular foundation model frameworks.

### 1.1. Related Work

*LLM-based table models.* Prior work serializes tables for LLMs (Hegselmann et al., 2023; Gardner et al., 2024), but is limited by context constraints and weak numerical reasoning (Bordt et al., 2023; Thawani et al., 2021).

*Numerical and in-context methods.* Models such as TabPFN, MITRA, and TabICL (Hollmann et al., 2025; Zhang et al., 2026; Qu et al., 2025; 2026) treat tables as numerical features, enabling scalability but ignoring semantic signals.

*Semantic-aware models.* Hybrid approaches (Spinaci et al., 2025; Kim et al., 2025; Chen et al., 2023) do incorporate semantic information but remain task-specific. More broadly, most models emphasize predictive performance

[1]IBM, San Jose CA, USA [2]TU Darmstadt, Germany [3]IBM, Yorktown Heights, USA. Correspondence to: Niharika S. D'Souza <Niharika.Dsouza@ibm.com>.

*Proceedings of the $2^{nd}$ ICML Workshop on Foundation Models for Structured Data*, Seoul, South Korea. 2026. Copyright 2026 by the author(s).

while general-purpose representations across table primitives remains under-explored.

## 2. HGR-TabE

Figure 1 first aligns heterogeneous (numeric and semantic) cell representations per row into a common space via an HGR objective (Sec. 2.3). The table is then represented as a hypergraph, where cells are nodes (with embeddings in the shared space) and hyperedges being rows, columns, table memberships. A Hypergraph Transformer (Sec. 2.4) produces structure-aware embeddings for all table elements.

### 2.1. Hypergraph Representation

A table $\mathcal{T} = [\mathbf{t}^H, \mathbf{C}, \mathbf{R}]$ with $N$ rows and $M$ columns is mapped to a hypergraph $\mathcal{G} = (\mathcal{V}, \mathcal{E})$, with incidence matrix $\mathcal{A} \in \{0,1\}^{mn \times (m+n+1)}$, where nodes $\mathcal{V}$ are cells $\mathbf{v}_{ij}$ and hyperedges $\mathcal{E} = \{\mathcal{R}_i, \mathcal{C}_j, \mathcal{T}^H\}$ encode row, column, and table membership. Node/hyperedge embeddings lie in $\mathcal{R}^D$.

### 2.2. Encoding Table Primitives

**Cell embeddings.** Each cell embedding $(\mathbf{x}_{ij})$ combines *context* (column header) and *content* to avoid ambiguity. Let $\mathbf{c}_j^H$ be the column header and $\mathbf{v}_{ij}$ be the content (for cell $i$) in column $j$. Non-numeric cells use sentence embeddings (Reimers & Gurevych, 2019; Wang et al., 2020): $\mathbf{x}_{ij}^s = [\mathrm{LM}(\mathbf{c}_j^H); \mathrm{LM}(\mathbf{v}_{ij})]$. Numeric cells use TabICLv2 (Qu et al., 2026): $\mathbf{W}_{\mathrm{TF}}, \mathbf{B}_{\mathrm{TF}} = \mathrm{TF}_{\mathrm{col}}(\{\mathbf{v}_{ij}\})$, $\tilde{\mathbf{c}}_{:,j}^n = \mathbf{W}_{\mathrm{TF}} \odot \mathbf{v}_j + \mathbf{B}_{\mathrm{TF}}$, giving $\mathbf{x}_{ij}^n = [\mathrm{LM}(\mathbf{c}_j^H); \tilde{\mathbf{c}}_{ij}^n]$.

**Row, column, and table embeddings.** We construct identifier embeddings to inject higher-level semantics prior to message passing. A **column embedding** aggregates both schema and value distribution by encoding a serialized description of the form *"Col: $\mathbf{c}_j^H$ | $Val_1$|…|$Val_K$"*, i.e. $\mathbf{e}_j^c = \mathrm{LM}(\text{col-desc}_j)$, where values are sampled or deduplicated to capture column semantics. A **row embedding** captures cross-column interactions via serialization $\mathbf{e}_i^r = \mathrm{LM}(\text{"}| \mathbf{v}_{i1} | \ldots | \mathbf{v}_{iM} |\text{"})$, preserving co-occurrence structure across heterogeneous features. A **table embedding** $\mathbf{e}^t = \mathrm{LM}(\mathbf{t}^H)$ (or concatenated headers if absent) provides global context. These are projected via an $\mathrm{MLP}_e(\cdot)$ to the same representation space $\mathcal{R}^D$ as cell embeddings during message passing (hypergraph representation learning).

### 2.3. Common-Space Alignment via soft-HGR

To align heterogenous cell modalities (numeric $\mathcal{N}$ vs semantic $\mathcal{S}$ columns), we learn modality-specific projections $\mathbf{f}^n(\cdot)$ and $\mathbf{f}^s(\cdot)$ that map numeric embeddings $\mathbf{x}_{in}^n, n \in \mathcal{N}$ and semantic embeddings $\mathbf{x}_{is}, s \in \mathcal{S}$ respectively into a shared latent space. $\mathbf{f}^n(\cdot)$ and $\mathbf{f}^s(\cdot)$ are designed as Set Transformers (Lee et al., 2019) with Induced Self Attention

Blocks (ISAB -See Fig. 1) to exploit column distributional properties and manage memory. We exploit the fact that cells co-occurring within the same row are statistically dependent, even across modalities, with dependence modelled using the Hirschfeld–Gebelein–Rényi (HGR) maximal correlation (Wang et al., 2019). Eq. 2.3 enforces the strongest possible nonlinear correlation between two random variables after optimal transformations under strict whitening constraints. Since directly enforcing whitening constraints is computationally expensive, we adopt the scalable soft-HGR loss approximation (Wang et al., 2019), replacing hard constraints with a covariance regularizer

$$\rho_{\mathrm{HGR}} = \sup \sum_{n \in \mathcal{N}, s \in \mathcal{S}} \left[ \mathbb{E}[\mathbf{f}^n(\mathbf{x}_{in})^\top \mathbf{f}^s(\mathbf{x}_{is})] \right] \quad (1)$$

$$\mathcal{L}_\rho = -\frac{\sum}{N_z} \left[ \mathbb{E}[\mathbf{f}^n(\mathbf{x}_{in})^\top \mathbf{f}^s(\mathbf{x}_{is})] + \frac{1}{2} \mathrm{Tr}[\mathrm{Cov}(\mathbf{f}^n)\mathrm{Cov}(\mathbf{f}^s)] \right]$$

where $N_z$ is the number of cross-modal pairs in a batch. The first term encourages aligned representations for co-occurring cells, while the second term prevents degenerate solutions by promoting decorrelated features. To stabilize training and preserve input information, we incorporate a reconstruction objective using modality-specific decoders, yielding the combined loss with $\lambda = 0.9$: $\mathcal{L} = \lambda \mathcal{L}_\rho + (1 - \lambda)\mathcal{L}_{\mathrm{MSE}}(\{\mathbf{x}_{ij}\})$. This balances cross-modal alignment with fidelity to the original representations. The resulting aligned embeddings $\tilde{\mathbf{v}}_{ij} = \mathbf{f}^n(\mathbf{x}_{ij}^n), \mathbf{f}^s(\mathbf{x}_{ij}^s) \in \mathcal{R}^{1024}$ are directly comparable for joint processing.

### 2.4. Hypergraph Transformer

Tables exhibit higher-order relational structure: cells are simultaneously grouped by rows, columns, and the table, inducing multi-way interactions that cannot be captured by pairwise graphs or sequence models without loss of information and/or obscuring the inherent permutation invariance across rows and columns. We circumvent this issue by using hypergraph transformers (Chen et al., 2023) for joint modelling of cells (nodes) and table primitives (hyperedges).

Given cell/hyperedge embeddings $\mathbf{H}_v^{(0)} = \{\tilde{\mathbf{v}}_{ij}\}, \mathbf{H}_e^{(0)} = \{\mathrm{MLP}_e(\mathbf{e}^{t/c/r})\}$, node-to-edge aggregation computes $\mathbf{Z}_e^{(l+1)} = \Phi_R(g_{\mathcal{V} \to \mathcal{E}}(\mathcal{A}_{:,e}, \mathbf{H}_v^{(l)}, \mathbf{H}_e^{(l)}))$. Each hyperedge aggregates information from all incident cells. This is followed by edge-to-node updates $\mathbf{H}_v^{(l+1)} = \mathrm{LN}(\Phi_R(g_{\mathcal{E} \to \mathcal{V}}(\mathcal{A}_{v,:}, \mathbf{Z}_e^{(l+1)})) + \mathbf{H}_v^{(l)})$, which provide context from row, column, and table- cell memberships. Attention is implemented via permutation-invariant hypergraph set attention (Bai et al., 2021), with $\tilde{\mathbf{I}} = \mathrm{MHA}(\mathbf{I})$ and $g(\mathbf{I}) = \mathrm{LN}(\tilde{\mathbf{I}} + \mathrm{rFFN}(\tilde{\mathbf{I}}))$ with multiheaded set attention (MHA), layer normalisation (LN), and Feed Forward (rFFN) modules. This formulation ensures that representations depend only on set membership rather than ordering. We stack $L = 12$ layers to enable iterative refinement of represen-

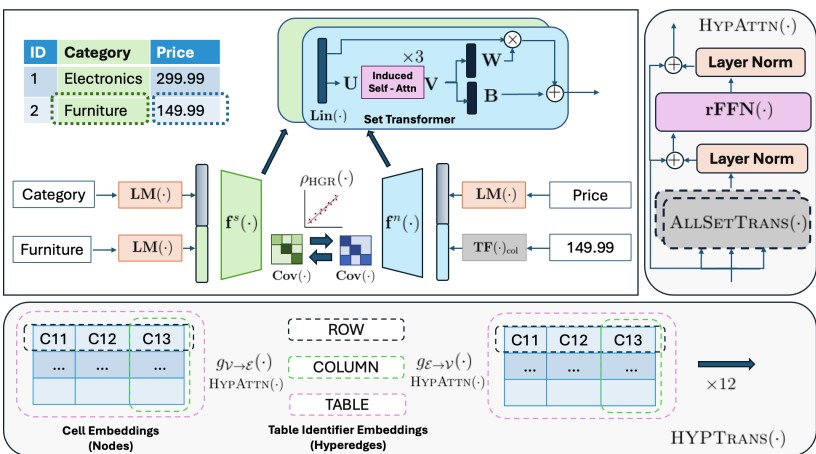

*Figure 1.* Illustrating our **HGR-TabE** framework. *(Top Left)* Common-space alignment of input tables. Row-wise cell values are encoded by type—LMs for non-numeric/semantic data (projected via $\mathbf{f}^s(\cdot)$) and TabICLv2's column encoder for numeric data (projected using $\mathbf{f}^n(\cdot)$) to maximize HGR correlation. *(Bottom)* A hypergraph transformer produces context-aware embeddings for cells, rows, columns, and table headers. *(Top Right)* The AllSetTransformer architecture that enforces structural invariance to row and column permutations.

tations across different granularities. Overall, this allows HGR-TabE to jointly model cell content and table structure, capturing both local interactions (within rows/columns) and global context (from the table) in a unified, permutation-invariant fashion.

Unlike (Chen et al., 2023), we explicitly fuse initial and contextualized embeddings at the final layer to preserve both local semantics and global structure.

$$\mathbf{Z}_v^{(L)} = \mathrm{mlp}_{\mathrm{fs}}([\mathbf{H}_v^{(L)}; \mathbf{H}_v^{(0)}]), \quad \mathbf{Z}_e^{(L)} = \mathrm{mlp}_{\mathrm{ft}}([\mathbf{H}_e^{(L)}; \mathbf{H}_e^{(0)}]).$$

This residual fusion MLPs are trained with dropout to mitigate over-smoothing and degeneracy in solutions. Thus, fine-grained cell content (from HGR-aligned embeddings) is retained alongside higher-order relational signals.

**Objective Function.** HGR-TabE uses a combination of reconstruction and contrastive objectives operating at different granularities of the table. These losses are designed to (i) preserve cell-level semantics, (ii) anchor higher-level representations to meaningful signals, and (iii) enforce invariance across views of the same table. The overall objective combines these components: $\mathcal{L} = \mathcal{L}_{\mathrm{cell}} + \lambda_{\mathrm{hyper}}\mathcal{L}_{\mathrm{hyper}} + \lambda_{\mathrm{InfoNCE}}\mathcal{L}_{\mathrm{InfoNCE}}$. Together, these losses balance local fidelity, global consistency, and cross-view invariance, enabling HGR-TabE to learn representations that generalize across tasks and tables ($\lambda_{\mathrm{hyper}} = 0.5$ and $\lambda_{\mathrm{InfoNCE}} = 0.03$)

**Cell reconstruction.** To ensure that the learned embeddings retain fine-grained content information, we apply masked reconstruction at the cell level. A fraction of cells is randomly masked, and the model is trained to reconstruct their original embeddings. We use separate decoders for numeric and non-numeric cells to respect modality differences. The loss is $\mathcal{L}_{\mathrm{cell}} = \frac{n_{\mathrm{num}}\mathcal{L}_{\mathrm{num}} + n_{\mathrm{cat}}\mathcal{L}_{\mathrm{cat}}}{n_{\mathrm{num}} + n_{\mathrm{cat}}}$, where

$\mathcal{L}_{\mathrm{num}} = \frac{1}{|\mathcal{M}_{\mathrm{num}}|} \sum_{v \in \mathcal{M}_{\mathrm{num}}} \|\mathcal{H}_{\mathrm{num}}(\mathbf{H}_{:,j}^{(L)}) - \mathbf{x}_{:,j}\|_2^2$ $(j \in \mathcal{N})$ and $\mathcal{L}_{\mathrm{cat}} = \frac{1}{|\mathcal{M}_{\mathrm{cat}}|} \sum_{v \in \mathcal{M}_{\mathrm{cat}}} \|\mathcal{H}_{\mathrm{cat}}(\mathbf{H}_{:,j}^{(L)}) - \mathbf{x}_{:,j}\|_2^2$ $(j \in \mathcal{S})$. This objective prevents representation collapse and information loss while encouraging faithful encoding of both numeric distributions and semantic content.

**Hyperedge reconstruction.** While cell reconstruction preserves local information, it does not directly constrain higher-level representations. To address this, we introduce a hyperedge reconstruction loss $\mathcal{L}_{\mathrm{hyper}} = \frac{1}{|\mathcal{E}|} \sum_{e \in \mathcal{E}} \|\mathcal{H}_e(\mathbf{H}_e^{(L)}) - \mathbf{e}\|_2^2$, which anchors row, column, and table embeddings to their initial semantic representations (derived from text encoders). This stabilizes training and ensures that global representations remain interpretable and aligned with input semantics.

**Contrastive consistency (InfoNCE).** To encourage invariance to stochastic corruption and improve generalization, we apply a contrastive loss on column and table embeddings. Two views ($w$) of the same table are generated via independent masking and dropout, and corresponding column/table embeddings are treated as positive pairs. The loss is $\mathcal{L}_{\mathrm{InfoNCE}} = -\frac{1}{K} \sum_k \log \frac{\exp(\mathbf{z}_k^{(1)} \cdot \mathbf{z}_k^{(2)}/\tau)}{\sum_j \exp(\mathbf{z}_k^{(1)} \cdot \mathbf{z}_j/\tau)}$, where $\mathbf{z}_k^{(w)}$ are normalized embeddings. This objective promotes robustness to missing data and encourages consistent representations across different views of the same underlying table.

## 3. Experiments

We train on tables from the CARTE dataset (Kim et al., 2024), providing rich table structures with semantic text and numerical columns with a total of over a million rows. We evaluate HGR-TabE on three tasks spanning different lev-

els of tabular representations: (i) tabular machine learning (classification/regression), (ii) row similarity search, and (iii) column similarity search. Row embeddings support both predictive modeling and instance-level retrieval (e.g., deduplication), while column embeddings enable schema-level tasks such as joinable column discovery. We compare against tabular foundation models (*Hytrel*, *TabICL*, *TabPFN*, *SAP-RPT-OSS*, *Tabula-8b*), and a general-purpose text embedding model (*all-MiniLM-L6-v2*), evaluating both predictive performance and embedding quality. TabPFN/TabuLA-8b do not support column embeddings. For TabICLv2, row, column embeddings are estimated using its individual transformer components before ICL.

### 3.1. Experiment Setup

**Row Similarity Search.** We evaluate on seven entity matching datasets (Köpcke et al., 2010; Mudgal et al., 2018) and two entity clustering datasets (Saeedi et al., 2017). Rows referring to the same entity define relevance. Each row is embedded and retrieval is performed via cosine similarity. We report Mean Reciprocal Rank (MRR), measuring how highly the first relevant match is ranked.

**Column Similarity Search.** We evaluate joinable column discovery on *Nextia* (Flores et al., 2021), *Valentine* (Koutras et al., 2021), *OpenData* (Kokel et al., 2025), and *WikiJoin-Small* (Srinivas et al., 2023). These asses the ability to identify semantically related columns across tables.

**Tabular Machine Learning.** We evaluate on 51 TabArena-Lite datasets (Erickson et al., 2025) using the standard "lite" split (fold 0), reporting ROC-AUC (binary), log-loss (multiclass), and RMSE (regression). For models without native prediction heads (e.g., HyTrel, MiniLM), we train downstream classifiers on frozen row embeddings and report on XGBoost, which performed best, following established protocols in tabular learning (Wu et al., 2024) and NLP (Li et al., 2020). For TabuLa-8B, we also use this pipeline due to unreliable label formatting in few-shot outputs. For HGR-TabE, we also evaluate multiple prediction heads and report Logistic Regression (one-vs-rest for multiclass and binned quantile classification with $N_q = 16$ for regression).

## 4. Results

Overall, no single model (Tab. 1) consistently outperforms others across all tasks, highlighting the trade-offs between general-purpose embeddings and task-specific tabular models. This underscores the need for a model capable of generalising across tabular tasks. Detailed results in Appendix B.

**Similarity Tasks.** On row similarity, HGR-TabE achieves a mean MRR of 0.50, outperforming tabular representation learning baselines such as HyTrel (0.05) and TabICL (0.04), while approaching the sentence-transformer MiniLM (0.66).

| Task →
Metric →
Method ↓ | Bin.
Class.
AUC
(↑) | Multi.
Class.
LogL.
(↓) | Reg.
-
RMSE
(↓) | Row
Sim
MRR
(↑) | Col
Sim.
MRR
(↑) |
|---|---|---|---|---|---|
| Hytrel | 0.50 | 2.81 | 38047.99 | 0.05 | 0.34 |
| SAP-RPT-OSS | 0.84 | 0.30 | 7179.57 | 0.03 | 0.08 |
| all-Mini-LM | 0.72 | 0.51 | 24117.74 | **0.66** | **0.59** |
| TabICLv2 | **0.86** | **0.24** | **6512.21** | 0.04 | 0.15 |
| TabPFNv2.5 | 0.84 | 0.27 | 6619.37 | 0.00 | - |
| Tabula8b | 0.78 | 0.44 | 15906.32 | 0.44 | - |
| **HGR-TabE** | 0.74 | 0.44 | 21991.60 | 0.50 | 0.54 |

*Table 1.* Results aggregated across datasets per task. ↑/↓ indicate whether higher or lower is better. HGR-TabE provides a balanced performance across tasks, though not as a clear winner. It outperforms specialised tabular FMs on similarity tasks, and general purpose embedding models on predictive tasks.

It attains second-best performance on several datasets, indicating robust instance-level representations. On column joins, HGR-TabE reaches a mean MRR of 0.54, outperforming tabular baselines and remaining competitive with MiniLM (0.59). These results suggest that HGR-TabE captures both row- and schema-level semantics effectively.

**Predictive Machine Learning.** HGR-TabE achieves a mean ROC-AUC of 0.74, outperforming HyTrel (0.50) and improving over MiniLM (0.72), while trailing specialized tabular models such as TabICL (0.86) and TabPFN (0.84). This reflects a common trade-off: methods optimized for predictive performance retain an advantage on supervised tasks. On multiclass classification, HGR-TabE achieves a mean log-loss of 0.44, improving over MiniLM (0.51) but trailing SAP-RPT (0.30), TabPFNv2.5 (0.27) and TabICLv2 (0.24). While not the top performer, it achieves competitive results and matches the best model on certain datasets. For regression, HGR-TabE achieves a mean RMSE of 21,991.6. This lags TabICL, SAP-RPT-OSS, and TabPFN, but outperforms HyTrel and MiniLM.

## 5. Conclusion

We present HGR-TabE, a unified tabular embedding model designed to generalize across diverse tasks without task-specific training. We show that: (i) HGR-TabE consistently improves over embedding models on predictive structured data tasks, (ii) it outperforms Tabular FMs on general table similarity tasks (iii) it provides a single representation that supports both predictive and similarity-based applications. At the same time, our findings highlight that no single model dominates across all tasks. Task-specific methods retain advantages in supervised prediction, while more general embedding models offer broader applicability. This suggests that HGR-TabE can serve as a flexible foundation, complementing rather than replacing specialized approaches.

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

## A. Dataset details

Table 2a and 2b shows the the statistics of the datasets used for row and column similarity. The number of testcases is the number of matches in the dataset.

| Dataset name | Type | # Rows | # Columns | # Testcases |
|---|---|---|---|---|
| Amazon-Google | EM | 4,589 | 4 | 234 |
| Beer | EM | 7,345 | 5 | 14 |
| DBLP-ACM | EM | 4,910 | 5 | 444 |
| DBLP-GoogleScholar | EM | 66,879 | 5 | 1,070 |
| Fodors-Zagats | EM | 864 | 7 | 22 |
| geological-settlements | C | 3,054 | 8 | 786 |
| iTunes-Amazon | EM | 62,830 | 9 | 27 |
| MusicBrainz | C | 19,375 | 10 | 5,000 |
| Walmart-Amazon | EM | 24,628 | 6 | 193 |

*(a)* Row Similarity Search Datasets. Type *EM* stands for datasets originally proposed for Entity Matching and *C* for Clustering. The number of testcases results from the number of positive matching pairs in each dataset.

| Dataset name | # Tables | # Rows | # Columns | # Testcases |
|---|---|---|---|---|
| nextia | 45+47 | 213,853 | 56.00 | 171+167 |
| valentine | 576 | 14,418 | 21.96 | 288 |
| opendata | 3102 | 1,742 | 8.58 | 42 |
| wikijoin-small | 659 | 44 | 2.54 | 100 |

*(b)* Column Similarity Search Datasets. We report the average number of rows and columns per dataset.

## B. Detailed Results Tables

In this section, we include the full result tables for all experiments from the main paper.

| Dataset | HyTrel | MiniLM | SAP-RPT-1 | TabICL v2 | TabPFN v2.5 | TabuLa-8B | HGR-TabE |
|---|---|---|---|---|---|---|---|
| Amazon-Google | 0.05 | **0.58** | 0.02 | 0.00 | 0.00 | 0.35 | 0.45 |
| Beer | 0.07 | **0.86** | 0.00 | 0.00 | 0.00 | 0.45 | 0.40 |
| DBLP-ACM | 0.16 | **0.96** | 0.02 | 0.00 | 0.00 | 0.95 | 0.96 |
| DBLP-GoogleScholar | 0.03 | **0.60** | - | 0.19 | - | 0.48 | 0.29 |
| Fodors-Zagats | 0.04 | **0.94** | 0.08 | 0.04 | 0.01 | 0.87 | 0.96 |
| MusicBrainz | 0.01 | **0.96** | 0.00 | 0.00 | 0.00 | 0.34 | 0.64 |
| Walmart-Amazon | 0.05 | **0.72** | 0.10 | 0.00 | 0.00 | 0.50 | 0.53 |
| geological-settlements | 0.00 | 0.06 | 0.00 | **0.15** | 0.00 | 0.00 | 0.13 |
| iTunes-Amazon | 0.00 | **0.23** | - | 0.00 | - | 0.03 | 0.16 |
| Mean | 0.05 | **0.66** | 0.03 | 0.04 | 0.00 | 0.44 | 0.50 |

*Table 3.* Row Similarity Search: Full Results MRR results per dataset. - indicates that the approach could not be run on the dataset, mostly due to memory constraints.

| Dataset | HyTrel | MiniLM | SAP-RPT-1 | TabICL v2 | HGR-TabE |
|---|---|---|---|---|---|
| Nextia | 0.25 | **0.26** | 0.01 | 0.08 | 0.24 |
| OpenData | 0.43 | **0.58** | 0.20 | 0.07 | 0.56 |
| Valentine | 0.26 | **0.59** | 0.00 | 0.42 | 0.50 |
| Wikijoin Small | 0.74 | **0.94** | 0.19 | 0.20 | 0.88 |
| Mean | 0.34 | **0.59** | 0.08 | 0.15 | 0.54 |

*Table 4.* Column Similarity Search. MRR results per dataset.

| Dataset | XGBoost | HyTrel* | MiniLM* | SAP-RPT | TabICL v2 | TabPFN v2.5 | TabuLa-8B* | HGR-TabE |
|---|---|---|---|---|---|---|---|---|
| APSFailure | 0.99 | - | 0.97 | 0.99 | **1.00** | - | 0.98 | 0.66 |
| Amazon_employee. | 0.82 | 0.50 | 0.72 | **0.85** | **0.85** | 0.84 | 0.73 | 0.68 |
| Bank_Customer_Chu. | 0.85 | 0.50 | 0.73 | 0.87 | **0.88** | **0.88** | 0.80 | 0.75 |
| Bioresponse | 0.88 | - | 0.60 | 0.87 | 0.87 | **0.89** | 0.78 | 0.71 |
| Diabetes130US | 0.63 | - | 0.60 | 0.65 | **0.68** | 0.67 | 0.61 | 0.65 |
| E-CommereShipping. | 0.74 | 0.51 | 0.67 | **0.75** | 0.74 | 0.74 | 0.70 | 0.71 |
| Fitness_Club | 0.76 | 0.53 | 0.68 | **0.81** | **0.81** | **0.81** | 0.77 | 0.74 |
| GiveMeSomeCredit | 0.86 | - | 0.77 | **0.87** | **0.87** | - | 0.84 | 0.85 |
| HR_Analytics_Job˙ | 0.80 | 0.51 | 0.77 | 0.81 | **0.82** | 0.81 | 0.78 | 0.80 |
| Is-this-a-good-cust. | 0.67 | 0.50 | 0.69 | **0.75** | 0.74 | 0.74 | 0.67 | 0.64 |
| Marketing_Campaign | 0.92 | 0.51 | 0.75 | 0.89 | **0.94** | **0.94** | 0.86 | 0.73 |
| NATICUSdroid | 0.98 | 0.48 | 0.88 | **0.99** | **0.99** | **0.99** | 0.96 | 0.96 |
| bank-marketing | 0.75 | 0.51 | 0.70 | 0.77 | **0.78** | 0.77 | 0.72 | 0.75 |
| blood-transfusion. | 0.65 | 0.45 | 0.62 | 0.68 | **0.74** | **0.74** | 0.69 | 0.67 |
| churn | 0.93 | 0.52 | 0.66 | 0.93 | **0.94** | **0.94** | 0.78 | 0.67 |
| coil2000_insurance. | 0.71 | - | 0.61 | 0.73 | **0.76** | **0.76** | 0.65 | 0.71 |
| credit-g | 0.77 | 0.53 | 0.74 | **0.80** | 0.79 | 0.79 | 0.75 | 0.76 |
| credit_card_clien. | 0.77 | 0.49 | 0.67 | **0.79** | **0.79** | **0.79** | 0.75 | 0.73 |
| customer_satisfact. | 0.99 | - | 0.92 | 0.99 | **1.00** | - | 0.99 | 0.94 |
| diabetes | 0.81 | 0.44 | 0.70 | 0.84 | **0.85** | **0.85** | 0.80 | 0.72 |
| hazelnut-spread. | 0.97 | 0.48 | 0.75 | **0.99** | **0.99** | **0.99** | 0.82 | 0.81 |
| heloc | 0.77 | 0.51 | 0.72 | **0.80** | **0.80** | **0.80** | 0.76 | 0.75 |
| in_vehicle_coupon. | 0.83 | 0.49 | 0.77 | 0.77 | **0.85** | **0.85** | 0.78 | 0.76 |
| jm1 | 0.73 | 0.51 | 0.70 | 0.75 | **0.79** | 0.77 | 0.72 | 0.70 |
| kddcup09_appetency | 0.77 | - | 0.55 | 0.81 | 0.81 | **0.82** | 0.56 | 0.70 |
| online_shoppers. | 0.92 | 0.49 | 0.86 | **0.93** | **0.93** | **0.93** | 0.89 | 0.85 |
| polish_companies˙ | 0.96 | 0.51 | 0.71 | 0.97 | 0.95 | **0.98** | 0.88 | 0.76 |
| qsar-biodeg | 0.91 | 0.47 | 0.77 | **0.94** | **0.94** | **0.94** | 0.90 | 0.78 |
| seismic-bumps | 0.74 | 0.47 | 0.66 | 0.78 | **0.81** | 0.79 | 0.75 | 0.68 |
| taiwanese_bankrupt. | 0.94 | 0.53 | 0.65 | 0.94 | **0.95** | **0.95** | 0.84 | 0.77 |
| Mean | 0.83 | 0.50 | 0.72 | 0.84 | **0.86** | 0.84 | 0.78 | 0.74 |

*Table 5.* Tabular Prediction : Binary Classification Results per Dataset measured by ROC AUC Score (higher is better). * indicates that the results were obtained by training XGBoost on top of row embeddings.

| Dataset | XGBoost | HyTrel* | MiniLM* | SAP-RPT | TabICL v2 | TabPFN v2.5 | TabuLa-8B* | HGR-TabE |
|---|---|---|---|---|---|---|---|---|
| MIC | 0.64 | 1.71 | 0.97 | **0.46** | 0.47 | **0.46** | 0.79 | 0.78 |
| SDSS17 | 0.09 | - | 0.33 | 0.08 | **0.07** | - | 0.13 | 0.30 |
| anneal | 0.04 | 2.58 | 0.17 | **0.02** | 0.03 | **0.02** | 0.08 | 0.10 |
| hiva_agnostic | 0.27 | - | 0.21 | **0.18** | 0.19 | **0.18** | 0.27 | **0.18** |
| maternal_health. | 0.41 | 3.49 | 0.40 | 0.50 | **0.36** | 0.42 | 0.43 | 0.56 |
| splice | 0.10 | 2.48 | 0.49 | 0.09 | **0.07** | 0.08 | 0.42 | 0.47 |
| students_dropout˙ | 0.64 | 2.96 | 1.07 | 0.55 | 0.55 | **0.54** | 0.93 | 0.82 |
| website_phishing | 0.31 | 3.64 | 0.46 | 0.54 | **0.20** | 0.21 | 0.43 | 0.32 |
| Mean | 0.31 | 2.81 | 0.51 | 0.30 | **0.24** | 0.27 | 0.44 | 0.44 |

*Table 6.* Tabular Prediction): Multiclass classification results measured by log loss (lower is better). * indicates that the results were obtained by training XGBoost on top of row embeddings.

| Dataset | XGBoost | HyTrel* | MiniLM* | SAP-RPT | TabICL-v2 | TabPFN-v2.5 | TabuLa-8B* | HGR-TabE |
|---|---|---|---|---|---|---|---|---|
| Another-Dataset-on. | 785.58 | 2357.67 | 1453.25 | 701.13 | 684.34 | **683.78** | 1084.08 | 1359.73 |
| Food_Delivery_Tim. | 7.61 | 10.65 | 9.01 | 7.98 | 7.66 | **7.60** | 8.25 | 8.54 |
| QSAR-TID-11 | 0.90 | - | 1.48 | 0.90 | 0.87 | **0.83** | 1.40 | 1.36 |
| QSAR_fish_toxicit. | 0.98 | 1.75 | 1.30 | 0.90 | **0.89** | 0.90 | 1.10 | 1.41 |
| airfoil_self_nois. | 1.59 | 8.93 | 5.29 | 1.17 | 0.99 | **0.97** | 3.70 | 4.71 |
| concrete_compress. | 4.80 | 22.25 | 12.84 | 4.05 | **3.66** | 3.82 | 9.79 | 11.41 |
| diamonds | 568.03 | 5441.49 | 1319.28 | 574.47 | 503.70 | **503.14** | 761.96 | 1154.98 |
| healthcare_insur. | 5016.73 | 15026.74 | 6698.41 | 4173.69 | **4053.12** | 4083.82 | 5739.85 | 6054.10 |
| houses | 0.23 | 0.70 | 0.48 | **0.20** | **0.20** | **0.20** | 0.34 | 0.44 |
| miami_housing | 96721.82 | 395649.83 | 304003.34 | 87855.92 | **79390.80** | 80753.88 | 199150.39 | 277275.23 |
| physiochemical. | 3.89 | 6.79 | 5.99 | 3.53 | **3.01** | 3.05 | 5.18 | 5.94 |
| superconductivity | 9.90 | - | 19.25 | 9.87 | **8.92** | 9.20 | 15.39 | 12.19 |
| wine_quality | 0.65 | 1.12 | 0.74 | 0.60 | **0.59** | 0.62 | 0.71 | 1.24 |
| Mean | 7932.52 | 38047.99 | 24117.74 | 7179.57 | **6512.21** | 6619.37 | 15906.32 | 21,991.6 |

*Table 7.* Tabular Prediction : Regression results per dataset measured by RMSE (lower is better). * indicates that the results were obtained by training XGBoost on top of row embeddings.

