# OpenReview forum: "HGR-TabE: Universal Tabular Embeddings via Maximal Correlation Alignment"
_ICML.cc/2026/Workshop/FMSD — FMSD @ ICML 2026 Poster_

### Official Review · Reviewer_qhQw · 2026-05-17
**Relevant universal tabular embedding direction, but empirical evidence is not yet convincing**

**Rating:** 5
**Confidence:** 3

**Review:**

**Summary:**

The paper proposes HGR-TabE, a self-supervised model for universal tabular embeddings. It aligns numeric and semantic cell representations using HGR maximal correlation and then applies a hypergraph transformer to produce cell, row, column, and table embeddings. The model is evaluated on predictive tasks, row similarity, and column similarity.

**Strengths:**

The problem is highly relevant to the workshop, since reusable tabular embeddings are an important direction for foundation models on structured data. The method is ambitious and attempts to handle multiple granularities of table representation. Evaluating row similarity, column similarity, and prediction is also a good step toward testing general-purpose tabular embeddings.

**Weaknesses:**

The empirical evidence does not fully support the strong “universal embedding” claim. HGR-TabE is competitive on similarity tasks, but it does not outperform MiniLM on average row/column similarity, and it trails specialized tabular models such as TabICL/TabPFN/SAP-RPT on prediction. The method is also quite complex, but there are limited ablations showing the contribution of HGR alignment, hypergraph message passing, reconstruction losses, and contrastive losses. Training only on CARTE tables may also limit the generality of the conclusions.

**Detailed comments:**

The paper should add ablations for each major component, stronger analysis of when HGR-TabE helps or fails, and clearer discussion of computational cost. The predictive evaluation should also include stronger downstream protocols, since the current results are weaker than standard tabular baselines. The related work could mention recent embedding-based table representation methods such as TabEmb and TabEmbed, which are closely related to reusable LLM/tabular embeddings.

**Justification:**

This is a relevant and promising direction, but the current paper overclaims relative to the results. I would support it more strongly if the authors provided clearer ablations and stronger evidence that the learned embeddings generalize across tasks better than simpler or existing alternatives.

---

### Official Review · Reviewer_SsGF · 2026-05-20
**HGR-TabE: a graph approach to obtain cell, rows, columns, and table embeddings**

**Rating:** 8
**Confidence:** 4

**Review:**

This paper proposes a model, HGR-TabE, to perform embeddings of tabular data at different levels (cells, rows, columns, and whole table). The model is trained in a self-supervised manner using multiple tasks and loss functions. Evaluation is performed on different downstream tasks (classification and regression using a shallow learner on top of the row embeddings; row similarity search again using row embeddings; column similarity search with column embeddings).

The model architecture is complex, reflecting the multitude of tasks that are asked of it. First, raw cell data is embedded in high dimension, concatenating text embeddings of headers and embeddings of contents depending on type (following ConTextTab and TabICL v2 methodologies). These are then mapped to a representation space in D dimensions via learned mappings f (implemented as ISAB blocks, again similar to TabICL). These maps are subject to a first loss, pulling together values appearing in the same row, with a regularizer given by enforcing non-vanishing variance (to avoid degenerate solutions). The resulting embeddings (one vector in R^D for each cell) are used as nodes of a hypergraph.

The hyperedges of the hypergraph correspond to joining cells that share a row, a column, or the table itself. The embedding of each row/column in the same R^D space is initialized as language model embeddings of the concatenation of all the values in that row or column (for columns: the column header is also added, and values are sampled/deduplicated as needed). For the table embedding, a description is used if present, otherwise the concatenation of the headers.

The embeddings above are then processed via a "hypergraph transformer." This allows refining the values of cells using values from all nodes connected to them, and then refining nodes using values of all edges departing from them. The final embeddings are obtained as an MLP applied to the concatenation of (1) values refined through this alternating attention procedure and (2) the original embeddings before the graph transformer.

During training, a number of reconstruction losses are applied: direct cell reconstruction of masked cells via L2 loss of their embeddings; hyperedge reconstruction; and contrastive consistency to ensure that views from the same table are embedded more similarly than views from different tables. Training is performed on the CARTE dataset, and evaluation happens on TabArena (for classification and regression) and on multiple matching and joinable column discovery benchmarks for row and column similarity tasks.

Final evaluation scores are mixed: the proposed HGR-TabE never ranks first in any task, while providing solid overall performance.

This paper shows a number of strengths, and in my opinion should be presented at the FMSD workshop:
- The overall approach is interesting, and clearly a lot of research went into the writing.
- Every piece of the architecture looks thought through, and I see no obvious improvements.
- The presentation has received great care, condensing a lot of information effectively into four pages.

At the same time, I see some weaknesses that prevent an outstanding score. Many of these are beyond the scope of a workshop paper and would rather need to be improved upon in view of future full publication. Some, however, are more fundamental issues.

Presentation (few and relatively minor issues):
- The main Figure 1 could be made more informative. It doesn't strike the right level of detail—for example, giving details of the inner workings of ISAB blocks while not making clear the actual architecture of HGR-TabE. Also, the print quality is low; a vector graphics export would look cleaner.
- The role of soft-HGR is not clear from a first reading. It appears that it is not optimized end-to-end with the other losses, but rather in a first step? Is training happening in a two-stage approach?
- In the description of the attention parts, there is some confusion between Z and H. I think in the first part of Section 2.4, it should always be $H_e^{(l+1)}$ and not $Z$, which should be reserved for the result of concatenation+MLP later.
- Some symbols could be described better. D becomes 1024 at some point, I think. And a $\mathcal{H_{...}}(H...)$ appears at some point without introduction (is it the reconstruction head, on top of Z? If so, it should not mention H but Z instead).
- The language throughout the paper is about hypergraphs, and claims to be built on top of the hypergraph attention of Bai et al. I'm not sure this is exactly what is being described, however. In the paper by Bai et al., hyperedges don't get their own embeddings and instead only guide the way in which the node embeddings are updated. The description in this paper (which is, in fact, possibly a more suitable approach than using Bai et al. here) appears to be a bipartite graph, with some of the nodes being cells and others being rows, columns, and the table itself—with some additional nuance because the weights used for "left to right" and "right to left" updates in the bipartite graph might differ. If so, it might be cleaner to present the paper in terms of a bipartite graph instead of talking about hypergraph attention.

Content (more important to address for a future paper submission):
- The initialization of the hyperedge embeddings looks hard to justify—a language model embedding of many consecutive values carries very little information. At the same time, there is no "obvious" other initialization that can be used, so it is a reasonable placeholder initialization value.
- More generally than the previous point, the main issue I have with this paper is that it builds a very large construction that cannot easily be ablated for testing smaller components. But still, for a full paper, many ablations would need to be run. Embeddings are constructed and aligned in a very non-trivial way. A large graph transformer is applied, only in the end to concatenate the results with the raw embeddings, making it unclear how much the graph was needed in the first place. Of course it's needed to some extent (e.g., to reconstruct masked cells), but it is less clear whether that really helps downstream.
- The pretraining should probably be pushed (much) further. The CARTE dataset is mostly used for downstream evaluation in other works because of its limited size. Using it as the sole resource for training seems unlikely to lead to any generalization power. Without knowing the full model details it is hard to tell how much data is needed, but quite likely more than this.
- Finally, the evaluation scores suggest that, unfortunately, some of the above concerns might be real. The paper is honestly framed as "an initial approach" since the abstract, so for a workshop this is clearly out of scope to demand more, but realistically the scores seem to suggest that this model adds very little on top of simply embedding values with all-MiniLM. Results are _worse_ than the simpler approach in both row and column similarity tasks, and just barely better in classification and regression. With these results, it is hard to justify using all the machinery developed here. Results might improve with more development, more training data, etc., but with more experimentation many pieces of the architecture might end up being changed too.
- Some pieces that might be worth checking for improvement are the loss functions. For reconstruction, squared L2 loss between embedding vectors might be too weak: the model might end up predicting some averaged scores (e.g., of the column distribution) instead of truly reconstructing the original value. Similarly, the contrastive loss might be too easy. Distinguishing, say, two columns from the same table vs. two columns from different tables is a very easy task, especially on such small-scale datasets. Distinguishing whether two tables are different views of the same table or not is essentially trivial. Also, the self-supervised task of cell reconstruction is basically the same as classification and regression tasks. It might be worth using this directly as a predictor, instead of stacking a logistic regression trained from scratch on top of the embeddings.

The above comments notwithstanding, I still judge this paper as a clear accept (score 8), because I believe that discussing it in the workshop setting can advance the general field of tabular deep learning.

---

### Official Review · Reviewer_N6Zu · 2026-05-22
**Intriguing Structural Approach, but Lacking Stronger Empirical Validation**

**Rating:** 5
**Confidence:** 4

**Review:**

### **Summary**
The paper introduces HGR-TabE, a self-supervised framework designed to generate universal tabular embeddings that support general tabular tasks, rather than being restricted to selective applications targeted by current TFMs (like pure classification/regression or isolated table understanding). The approach aligns separate numeric and semantic cell representations into a shared latent space using a soft-Hirschfeld-Gebelein-Rényi (HGR) maximal correlation objective. These aligned features are then processed by a 12-layer Hypergraph Transformer to model multi-way cell, row, and column interactions while preserving structural permutation invariance. The authors evaluate the framework across standard tabular prediction tasks (classification/regression) and similarity tasks (entity matching, column joins).

### **Strengths**
- **Novel Alignment Mechanism:** The application of the HGR maximal correlation constraint to bridge the modality gap between numerical values and semantic text within tabular rows is mathematically sound and a solid attempt at solving tabular heterogeneity.

- **Structural Preservation:** Framing the table as a hypergraph and utilizing an All-Set Hypergraph Transformer effectively respects the multi-way relational geometry of tabular data. This avoids the structural destruction caused by standard LLM linearization and offers an interesting alternative to related graph-based approaches like CARTE (Kim et al., 2024)[https://arxiv.org/pdf/2402.16785].

- **Ambitious Multi-Task Scope:** The attempt to unify instance-level retrieval (row similarity), schema-level matching (column similarity), and standard tabular ML under a single representation space is highly relevant to the workshop's theme of foundation models.

### **Areas for Improvement**
- **Possible Training/Eval Data Contamination:** The evaluation does not address the severe risk of test-data leakage between the CARTE corpus used for pre-training and the downstream evaluation benchmarks (like TabArena-Lite).

- **Missing Citations and Limitations:** While the framework has clear boundaries, the paper lacks a dedicated Limitations section. Furthermore, the related work is quite brief, prematurely dismissing semantic-aware models as "task-specific." For example, the CARTE framework shares a fundamental graph-based philosophy for dealing with heterogeneous schemas and entities. Ironically, while the CARTE dataset is explicitly used for training, the foundational paper itself is completely missing from the bibliography and related work.

- **Omitted Baselines:** The most relevant semantic tabular foundation models (e.g., CARTE, TARTE) are not included in the experiments. CARTE utilizes a graph-based structural representation, while TARTE leverages a knowledge-augmented Transformer architecture. Despite architectural differences, both are natively suited for heterogeneous text/numeric processing and share the exact data ecosystem as the proposed model. Theoretically, they would stand a much better chance in the similarity experiments than the chosen baselines, which scored literal zeros (e.g., TabPFN, TabICL).

- **Ambiguous Baseline Setup:** The paper tests prediction-focused models (like TabPFN and TabICLv2) on vector similarity tasks, but never explains how it extracted the embeddings/data to do so. This lack of transparency, coupled with staggeringly low baseline scores (0.00 MRR), calls into question the fairness and correctness of this experimental setup.

- **Semantic Capabilities Claim vs. Limited Evaluation:** While the paper explicitly deals with semantic understanding and text feature processing, the selected benchmarks do not rigorously evaluate this. Appropriate semantic benchmarks like TabFact or TextTabBench (provided they are rigorously sanitized against the pre-training set) would serve as much better stress tests for the model's text-feature processing. I understand this is a workshop paper, so this is more of a 'wish it was included' point for future iterations.

- **Somewhat Overextended Claims:** The claim of "universal" embeddings is heavily undermined by the restricted scope of the 2-3 evaluated task domains, underperformance on standard machine learning benchmarks compared to specialized models, and the complete absence of more complex/specialized semantic evaluations (e.g., Table QA, which was used to motivate the paper in the introduction).

### **Detailed Comments**
**1) Address Data Leakage:** Can the authors explicitly clarify whether any of the evaluation datasets (most importantly TabArena-Lite) were filtered out of the CARTE benchmark training corpus?

**2) Baseline Extraction:** Please document the exact layer-extraction and pooling protocol used to pull cosine-comparable embeddings from TabPFN and TabICL. As demonstrated in recent foundation model literature (e.g., Feofanov et al., arXiv:2602.17868), zero-shot capabilities across distinct tasks are highly sensitive to layer selection from the FM. Extracting raw final-layer representations from a predictive network often collapses performance on off-target tasks like vector retrieval. Without detailing this extraction protocol or validating intermediate-layer representations, the 0.00 scores appear too low without further investigation.

**3) Include Direct Competitors:** The inclusion of CARTE and/or TARTE as baselines for the row and column similarity tasks is strongly recommended for future iterations, as they are HGR-TabE's close ideological competitors.

**4) Fix Citations:** Please ensure the CARTE foundational paper (CARTE: Pretraining and Transfer for Tabular Learning, Kim et al., 2024) is properly cited where the dataset is introduced.

### **Justification of Score**
**Recommended Score: Weak Reject**
The paper introduces an intuitively appealing architecture that appropriately handles the multi-modal geometry of tabular data. However, the empirical execution contains fundamental flaws that, for me, prevent acceptance in its current state. The unaddressed risk of training/evaluation data leakage, undetailed baseline configurations yielding poor results, and missing the model's closest architectural competitors render the experimental results unclear. Furthermore, the paper overclaims "universality" while ignoring more complex semantic tasks and trailing standard models on basic predictive machine learning. While the methodology is promising and well-suited for the workshop theme, the evaluation design requires an overhaul to prove the model's true capabilities.